# Contamination and Health Risk Assessment of Heavy Metals in the Soil of Major Cities in Mongolia

**DOI:** 10.3390/ijerph16142552

**Published:** 2019-07-17

**Authors:** Sonomdagva Chonokhuu, Chultem Batbold, Byambatseren Chuluunpurev, Enkhchimeg Battsengel, Batsuren Dorjsuren, Batdelger Byambaa

**Affiliations:** 1Department of Environment and Forest Engineering, School of Engineering and Applied Science, National University of Mongolia, Ulaanbaatar 210646, Mongolia; 2Department of Transdisciplinary Science and Engineering, School of Environment and Society, Tokyo Institute of Technology, Tokyo 145-0066, Japan

**Keywords:** heavy metal, soil, contamination level, health risk assessment, Ulaanbaatar, Erdenet, Darkhan

## Abstract

Using the case of Ulaanbaatar, Erdenet, and Darkhan cities from Mongolia, the study aimed to assess the contamination level and health risk assessment of heavy metals (As, Cr, Pb, Ni, and Zn) in urban soil. A total of 78 samples was collected from a variety of functional areas. The geoaccumulation index (I_geo_) and integrated pollution index (IPI) were used in pollution assessment, while the health risk was scored using a hazard quotient (*HQ*) and health index (HI) for non-carcinogenic heavy metals, as well as a lifetime average daily dose (*LADD*) for carcinogenic heavy metals. The results show that the concentration of heavy metals in the soil samples taken from Darkhan city, which presented “uncontaminated” values in terms of I_geo_ for all metals, was relatively lower than other cities within the contamination assessment. Furthermore, the I_geo_ value signified “uncontimated to heavily contaminated” soil in the Ulaanbaatar and Erdenet cities. Typically, as for the IPI that observed similar trends with I_geo_, the mean IPI values in Ulaanbaatar, Erdenet, and Darkhan were 1.33 (moderate level of pollution), 1.83 (moderate level of pollution), and 0.94 (low level of pollution), respectively. In terms of the assessment of potential health risk, there was a particular or different level of ingestion, dermal contact, and inhalation exposure pathway for human health. Among these three different pathways, the ingestion was estimated by the main contributor for health risk. Each value of HQ and HI indicated that soil heavy metals of studied cities were at a safe level (<1) or had the absence of a significant health risk there. In addition, the potential health risk for children was greater than for adults, where heavy metal values of HI for children had a high value compared to adults. We estimated carcinogenic risks through the inhalation exposure, and as a result, there were no significant risks for human health in the studied cities from three elements (As, Cr, and Ni).

## 1. Introduction

During the last few decades, the properties of heavy metals in urban soils and street dust from urban roads have gained increasing interest, as indicated by the increasing number of papers [1]. This increase is probably attributed to the potential public health risk due to the intake of heavy metals. Heavy metals in urban soils and street dust from urban roads can accumulate in the human body via direct inhalation, ingestion, and dermal contact absorption [2,3,4,5,6,7,8,9]. According to numerous studies, the anthropogenic source of heavy metal pollution is considered to be the main factor for environmental pollution issues, especially in urban areas. The anthropogenic sources of heavy metals include traffic emission (fuel burning in the engine, vehicle exhaust particles, tire wear particles, weathered street surface particles), industrial emission (power plants, coal combustion, metallurgical industry, auto repair shop, chemical plant, etc.), domestic emission, weathering of buildings and pavement surfaces, deposited from the atmosphere, and so on [8,10]. For instance, traffic pollution emits Pb, Zn, and Cu, whereas Ni is associated with naturally occurring sources, Cd originates from industrial contaminants, and Cr is associated with atmospheric deposition [11]. Due to the large impact on the environment and human health, some heavy metals, such as Cd, Zn, and Pb, are a public concern [12,13]. Therefore, studying the chemical properties of urban soil gives the opportunity to evaluate the urban environment quality, as well as its relation to human health.

Many studies have been performed on heavy metal pollution of soil around the world [14,15,16]. During this time, quite a few studies have considered the concentration of heavy metal and its potential risk assessment for human health from urban soil. However, there are limited studies about heavy metals of soil in major municipalities in Mongolia [17,18,19]. Ulaanbaatar, Erdenet, and Darkhan are the centers of socio-economy, industrialization, and transportation in Mongolia. Thus, in comparison with major cities or settlement areas, the soil contamination and human health risks associated with heavy metals in urban and industrial areas require further investigation.

In Mongolia, several areas with relatively high population densities, including the capital city, are increasing land use and have developed a low efficiency manufacturing industry over the past few decades, and its impact may cause the issue of heavy metal pollution in the urban environment. With contamination level in its present condition, by using the geoaccumulation index and integrated pollution index, we indicate the soil heavy metal pollution in three cities in Mongolia. In order to evaluate the potential human health risk, we utilized the methodology, developed by the Environmental Protection Agency of United States (US EPA) and attempted to estimate non-carcinogenic and carcinogenic risk via five heavy metals (As, Cr, Ni, Zn, and Pb) concentration for children and adults, separately. The result of the health risk assessment gives a crucial understanding about the current situation and required regulations for urban planning to governments. This study aims to (1) determine the concentration of five different heavy metals, such as Cr, As, Ni, Zn, and Pb, and then to evaluate the heavy metal pollution using the geoaccumulation index and integrated pollution index; (2) to assess the non-carcinogenic and carcinogenic health risk for humans from five different metal elements; and (3) to compare the studied cities with each other (Ulaanbaatar, Erdenet, and Darkhan).

## 2. Materials and Methods 

### 2.1. Study Areas

Ulaanbaatar (UB) is the capital city of Mongolia (47°38′53″–48°16′19″ N, 106°21′46″–107°37′17″ E) which is located in the intermountain by the Tuul River at an altitude of about 1350 m above sea level. The winter climate is extremely continental with frequent temperature inversions. The mean annual precipitation in UB is 240–260 mm, where around 60–90% of the precipitation is in July and August. As of the classification of the soil geographical region and characteristics, this area is included in the Khentii subrange within the Khangai range and darkish-brown (Kastanozems) and mountain darkish-brown (Mollic Cambisols) soil cover most of the area [20,21] (Table 1).

Erdenet (49°07′54″–48°58′52″ N, 103°58′27″–104°11′59″ E), which is the largest industrial center of Mongolia with enrichment and primary processing of non-ferrous metals (copper and molybdenum), is located at an altitude of about 1200 m above sea level. In terms of the classification of the soil geographical region and soil characteristics, this area belongs to the Khangai subrange within the Khangai range and has mountain darkish-brown (Mollic Cambisols) soil [20]. 

Darkhan (49°32′04″–49°23′10″ N, 105°53′17″–106°1′13″ E), where the manufacturing sector is primarily metallurgic and light and has many other construction industries, is located at an altitude of about 650 m above sea level. Using the soil geographical classification, the city has soils in the darkish-brown/brown subrange within the Khangai range, and alluvial (Fluvisols) and brown (Aridic Kastanozems) soil dominate in this area [20].

The pollution sources of all three cities are associated with thermal power plants (TPP), industrial activities, mining operation, vehicles, and ger districts (Mongolian traditional or detached house that is not linked to infrastructure) [22].

### 2.2. Soil Sampling

A total of 78 soil samples was taken from a variety of functional zones (Figure 1); i.e., apartment district, ger district, industrial sites, and common area for three different cities. Within this study, the sampling processes were carried out between 2016 and 2018 in order to investigate the concentration and assess the contamination level of the heavy metals in those cities. Each sampling point was situated near the roadside or off-road sites. About 1 kg of surface soil (<10 cm depth) samples were collected from each area of a square meter measured by a ruler and gathered into a self-sealing polyethylene bag using a shovel. The exact location (longitudes and latitudes) of each sample point was determined using a global positioning system (GPS) instrument (Garmin Etrex 10 GPS, Garmin Ltd, Lenexa, Kansas, USA). Finally, the soil samples were transferred to the laboratory (Khanlab LLC, Ulaanbaatar, Mongolia) with the sample-box by cab for further analysis [23,24,25].

Prior to determination of the heavy metals concentrations, sieved soil samples had a mixture of HCl-saline, HNO_3_−, HClO_4_, and HF solution applied to them. Afterwards, the quantitative analyses of heavy metal elements were carried out by recording the intensity of the atomic radiant emissions at high temperature. A iCAP-7400 instrument (Thermo Fisher Scientific Inc., Waltham, Massachusetts, USA) was used in the analyzing process.

### 2.3. The Assessment of Heavy Metal Contamination

The geo-accumulation index (I_geo_), developed by Muller, has been used since the late 1960s and was previously widely adopted to assess the metal pollution in European trace metal studies [26]. The I_geo_ was used to assess heavy metal pollution in urban soils by comparing the present and background concentrations. I_geo_ was estimated using [27]:(1)$$I_{geo}=\log_{2}\left(\frac{Cn}{1.5Bn}\right)$$
where *C_n_* is the concentration of an element in urban soil and *B_n_* is the background value. In this study, the chemical composition of non-effected soil at a site of each city [18,23,28] was chosen as the background values for calculating the I_geo_ values. The constant 1.5 allowed us to analyze natural fluctuations in the concentration of a given substance in the environment and to detect very small anthropogenic influences [27].

The results based on the geo-accumulation index are divided into seven classes (Table 2). Class 7 is an open class and may be hundreds of times higher than the background value [27].

To further assess the contamination of urban soil, we used the pollution index (PI), which is estimated at each site by an element, and the integrated pollution index (IPI), which is represented by the mean value of IPs of the metals at different settlement areas. The PI of each metal was calculated as the ratio of the heavy metal concentration in the study to the background concentration of the corresponding metal, as follows [15,29]:(2)$$PI_{i}=\frac{C_{i}}{B_{I}}$$
where *C_i_* is the concentration of an element in the urban soil and *B_i_* is the background value of the study areas. Having calculated each element at each site, the IPs were averaged by sampling the sites using Equation (3). In order to determine the difference between settlement areas, we grouped those into four different (three for Darkhan) areas. The IPI is classified as IPI ≤ 1 for a low level of pollution, 1 < IPI ≤ 2 for a moderate level of pollution, 2 < IPI ≤ 5 for a high level of pollution, and IPI > 5 for an extreme high level of pollution [15,29]. 

(3)$$IPI_{i}=\left(IP_{1}+IP_{2}+IP_{3}+\cdots +IP_{n}\right)/n$$

### 2.4. Health Risk Assessment

#### 2.4.1. Exposure Dose

In this study, the risk assessment methodology introduced by the Environmental Protection Agency of the United States (US EPA) was used to evaluate the health risks in relation to the concentration of heavy metals in urban soil. According to the Exposure Factors Handbook, the average daily dose (*ADD*) (mg/kg/day) of three exposure pathways; ingestion, dermal contact, and inhalation can be estimated using Equations (4)–(6):(4)$$ADD_{ing}=\frac{c\times R_{ing}\times CF\times ED}{BW\times AT}$$
(5)$$ADD_{inh}=\frac{c\times R_{inh}\times EF\times ED}{PEF\times BW\times AT}$$
(6)$$ADD_{derm}=\frac{c\times SA\times CF\times SL\times ABS\times ED}{BW\times AT}$$
where *ADD_ing_* is daily exposure amount of metals through ingestion (mg/kg/day), *ADD_inh_* is daily exposure amount of metals through inhalation (mg/kg/day), and *ADDd_derm_* is daily exposure amount of metals through inhalation (mg/kg/day). The exposure factors for these models are shown in Table 3 with reference to the US EPA and environmental site assessment guidelines. The values of these factors include the standards from US EPA and concrete data from this study. 

#### 2.4.2. Non-Carcinogenic Risk Assessment

After the calculation of *ADD*, the three exposure pathways are calculated, a hazard quotient (HQ) based on the non-carcinogenic toxic risk can then be calculated by dividing the daily dose by a particular reference dose (*RfD*), as in Equation (7):(7)$$HQ=\frac{ADD}{RfD}$$

Using the threshold of *RfD* value, it is possible to evaluate whether there are existing adverse health effects to humans, and when *RfD* value is higher than the *ADD*, there would be not any adverse health effect [30]. If the *HQ* value indicates lower than 1, there are no adverse health effects; whereas, an *HQ* higher than 1 means there is likely adverse health effects [31].

The hazard index (HI) can then be calculated by adding *HQ*s of a mix of metal elements: (8)$$\mathrm{HI}=\overset{5}{\underset{i=1}{\sum }}HQ_{i}$$

HI is equal to the sum of the *HQ*s and gives the total risk of being non-carcinogenic for a single element. If the value of HI ≤ 1, it is considered that “no significant risk” of non-carcinogenic effects exists. However, when HI > 1, there is a probability of non-carcinogenic effects occurring, and the probability increases with a rising value of HI [32]. For the present study, HI will be utilized in assessing the human health risk of exposure from five heavy metal elements in urban soil.

#### 2.4.3. Carcinogenic Risk Assessment

For carcinogens, the lifetime average daily dose (*LADD*) (inhalation exposure route for As, Cr, and Ni) was utilized in the evaluation of carcinogenic risk, and according to the classification list developed by the International Agency for Research on Cancer (IARC), to investigate the three heavy metals for their carcinogenic risks [33,34,35]. The lifetime average daily dose for these three metals was calculated using: (9)$$LADD=\frac{C\;x\;EF}{AT}\times \left(\frac{CR_{child}\times ED_{child}}{BW_{child}}\times \frac{CR_{Adult}\times ED_{adult}}{BW_{adult}}\right)$$
where all the variables are the same as in Equations (4)–(6).

*CR* is the inhalation (*CR* = *R_inh_*) rate. The interpretations and values of other parameters are listed in Table 3. The lifetime cancer risk can be calculated using:(10)R=LADD/SF
where *SF* is the corresponding slope factor. The level of cancer risk associated with exposure to those elements in soil is the range of threshold values (10^−6^–10^−4^), above which environmental and regulatory agencies consider the unacceptable risk [35]. 

## 3. Results

The results section was presented using the geoaccumulation Index (I_geo_), the integrated pollution index (IPI), and the health risk assessment (non-cancer and cancer risk). In order to define the health risk assessment, the exposure doses were calculated using the concentration of heavy metals, and the non-cancer risk assessment result was revealed for each analyzed element and city. 

### 3.1. Heavy Metal Concentration

The concentration of five elements (As, Cr, Pb, Ni, and Zn) in Ulaanbaatar, Erdenet, and Darkhan with their background values were shown in Table 4. 

#### 3.1.1. Ulaanbaatar

As presented in Table 4, the mean concentrations of As, Cr, Pb, Ni, and Zn in urban soils were 28.04, 16.56, 43.11, 21.26, and 106.11 mg/kg, respectively. The concentration ranges of the metals were observed to be 7.0–68.0, 2.0–41.0, 11.0–166.0, 5.0–60.0, and 46.0–209.0 mg/kg, respectively. It is also significantly apparent that the mean concentrations of all elements in the soil from UB exceed their background values with the exception of Zn (106.11). However, arsenic (As) concentration was 4.7 times greater than maximum permissible concentration (MPC). 

#### 3.1.2. Erdenet

The mean concentrations of As (12.78), Cr (65.70), Pb (18.06), Ni (29.30), and Zn (155.17) mg/kg in urban soil of all the sites from Erdenet were from 1.2 to 3.2 times higher than their background values. The concentration ranges of the metals were observed to be 4.90–24.00 (As), 22.00–198.00 (Cr), 4.90–65.00 (Pb), 10.00–57.00 (Ni), and 42.00–914.00 (Zn) mg/kg. Whereas the mean concentrations of Cr, Pb, Ni, and Zn were much lower than their MPC, As was 2.13 times higher than the MPC (Table 4). 

#### 3.1.3. Darkhan

The mean concentrations of As (3.33 mg/kg), Cr (31.90 mg/kg), Pb (20.90 mg/kg), Ni (19.49 mg/kg), and Zn (67.26 mg/kg) in soils of the 21 sites from Darkhan city ranged from 1.93–5.58, 3.28–55.54, 9.45–66.35, 5.85–64.30, and 37.69–122.00 mg/kg, respectively. In addition, the mean concentrations of Cr and Pb were lower than their background values, while Ni and Zn were higher than their background values. Compared to MPC, each of the elements were less than their respective standard value (Table 4). 

### 3.2. Contamination Level of Heavy Metals

#### 3.2.1. Geo-Accumulation Index 

The minimum, maximum, and mean values of I_geo_ were shown in Table 5. I_geo_ values were calculated on each sampling point. The aim was to evaluate the overall contamination levels of the sampled cities. The I_geo_ data and the Muller’s geoaccumulation index was listed in Table 2. 

##### Ulaanbaatar

As shown in Table 5 and Figure 2, the mean I_geo_ values were in the range from −0.78 to −0.21, indicating that soil in Ulaanbaatar city was uncontaminated with those elements (Cr, Pb, Ni, and Zn) except for As (0.26), which showed a moderately contaminated value using the I_geo_ estimation. The highest I_geo_ values for As, Cr, Pb, and Ni also showed that the soil was moderately contaminated, whereas the Zn (0.29) value indicated uncontaminated to moderately contaminated soil. In terms of the highest I_geo_ value, some sampling sites were slightly impacted by anthropogenic factors. As, Cr, Pb, and Ni were 1.68, 1.07, 1.55, and 1.44, respectively.

##### Erdenet

The main mean I_geo_ values for Cr, Pb, Ni, and Zn were lower than 0 or uncontaminated, while As was 0.84. The highest I_geo_ values for each element were higher than 1. For instance, As and Zn showed the highest I_geo_ values ranged from 2 to 2.97, which indicated moderately to heavily contaminated; Cr, Pb, and Ni were also 1.14, 1.53, and 1.03, respectively (Table 5 and Figure 2). 

##### Darkhan

As presented in Table 5 and Figure 2, the main mean values for each element were lower than 0. This means that the soil of Darkhan city was uncontaminated by all elements used for this study. Whereas the highest value is estimated 0.16 (As), 0.64 (Ni), and 1.75 (Zn), which indicated uncontaminated to moderately contaminated soil with respect to them. Rest of elements, Cr and Pb are −0.04 and −0.08, respectively. 

#### 3.2.2. Heavy Metal Pollution Index (PI)

Table 6 and Figure 3 represents the IPIs (Integrated Pollution Indices) for different functional areas from the cities regarding the five elements of As, Cr, Pb, Ni, and Zn. In terms of Ulaanbaatar and Erdenet city, integrated pollution indices were higher than Darkhan city (as shown in Figure 3); for instance, the mean IPI values in Ulaanbaatar, Erdenet, and Darkhan were 1.33 (moderate level of pollution), 1.83 (moderate level of pollution), and 0.94 (low level of pollution), respectively. 

Regarding the pollution index, there were some differences in settlement areas depending on their socio-economic features, aging of cities, and so on. Integrated pollution level for functional areas in cities decreased in the following order:Ulaanbaatar: Industrial area > ger district > apartment districtErdenet: Industrial area > ger district > apartment district > common areaDarkhan: Industrial area > apartment district > ger district > common area

Among these sites, the industrial areas are more polluted than in other functional areas in the studied cities (Table 6).

### 3.3. Health Risk Assessment

#### 3.3.1. Non-Carcinogenic Risk

The reference dose (*RfD*), upper confidence level (95% UCL), and results of the *HQ* and HI calculations from three cities are listed in Table 7. The HQ and HI values for both children and adults have the same patterns. The *HQ* of the ingestion of soil particles for all metals was much higher than those of the inhalation of re-suspended soil particles and dermal absorption with soil particles. *HQ* of ingestion of soil particles for all metals was much higher than those of the inhalation and dermal absorption exposure pathways. The values of *HQ* and HI for those pathways of this study decreased in the order of ingestion > dermal contact > inhalation. The contribution of *HQ_ing_* to HI was the highest in exposure pathways and accounted for more than 85% of the total risk. This result shows that the ingestion is a principal pathway of heavy metals damaging to human health. Conversely, the inhalation and the dermal contact had relatively low values. 

The non-cancer risk of As was a major contribution to HI (from 2.04 × 10^−4^ to 4.15 × 10^−3^) was lower than the safe level. Therefore, the potential health risks for children and adults can be overlooked. Meanwhile, the health risk for adults was lower compared to the risk for children. The HI values of these metals for children were from 2 to 9 times higher than those for adults.

#### 3.3.2. Carcinogenic Risk

The carcinogenic risks according to inhalation exposure to As, Cr, Pb, and Ni are presented in Table 8. Results showed that the risks for As, Cr, and Ni decreased in the sequence of Ni > As >> Cr for UB, whereas it was in the order of Ni > Cr > As for Erdenet and Darkhan. Levels of carcinogenic risks for those elements were lower than the tolerable range (10^−6^–10^−4^), above which the environmental and regulatory agencies perceive as an unacceptable risk. 

## 4. Discussion

### 4.1. Heavy Metal Concentration

Depending on the geochemical properties of the soil, the concentration of elements decreased in the following order; As > Cr > Ni > Pb > Zn for each city. The heavy metals concentration of Darkhan city’s soil was lower than Ulaanbaatar and Erdenet cities; for example, As (3.9–8.5 times greater than Darkhan) and Zn (1.6–2.3 times greater than Darkhan) were much high in UB and Erdenet. This pattern was also observed in the background values of cities. All elements used for this study, except for As, indicated values much lower than their respective MPC value (Table 4). Since this MPC is evolved from geochemical studies and comprehensive characteristic of each soil type in Mongolia, this standard probably may be considered to be general values in contamination assessment. Collating the concentration of heavy metals in local area soils with the MPC, it might be inaccurate to determine whether the soil is contaminated or not. Therefore, the background values were identified and used for the assessment. Arsenic is a ubiquitous element that is detected at low concentrations in virtually all environmental matrices. Natural levels of arsenic in soil usually range from 1 to 40 mg/kg [39]. Therefore, it is considered to be normal in terms of concentration for each city.

### 4.2. Heavy Metal Contamination Assessment

In terms of the assessment based on the background value, mean values of I_geo_ indicates an “uncontaminated” level in each city, apart from arsenic, which demonstrated an “uncontaminated to moderately contaminated” value for Erdenet and Ulaanbaatar. In Ulaanbaatar, the I_geo_ value of all elements showed similar trends and their fluctuation was not extremely high. Regarding Erdenet, I_geo_ values for As and Zn were higher than for Cr, Pb, and Ni. Namely, Zn ranged from −1.47 to 2.97, and at its maximum value, reached the “moderately to heavily contaminated” soil criterion. Soil contains zinc at concentrations of 5–770 mg/kg with an average of 64 mg/kg worldwide [40], while it is 300 mg/kg in soil according to the Mongolian MPC [38]. This element is normally found in association with other base metals such as copper and lead in ores [41]. Erdenet city has been used since 1978 as the largest mining operation area and Erdenet Mining Corporation is considered one of the biggest ore mining and ore processing factory in Asia. The mine yields approximately 354 thousand metric tons of copper concentrates and 3500 metric tons of molybdenum concentrate per year [42]. Therefore, the level of heavy metal contamination in Erdenet might be totally caused by human activities or mining.

Erdenet is a long-term mining site (since 1978), and 26.7 percent of the total samples were taken from the mine site. This may have contributed to the high pollution level. Darkhan is regarded as an industrial center; however, the population, the number of households living in ger districts, and the number of vehicles that are considered the main pollutant source is less than the other two cities. However, sampling from Ulaanbaatar did not involve large-scale industrial sites, where the integrated pollution index is at 2.15, or high-level pollution. This may be due to the fact that almost half the population of Mongolia live there. As a result, other pollution sources increase the major pollution concern. In addition, Ulaanbaatar is the oldest city of Mongolia (Table 1).

### 4.3. Health Risk Assessment

*HQ*s and HIs of the five metals were all much lower than the safe level (=1), indicating that the adverse health impact on children’s and adults’ exposure to heavy metals in soil were relatively light in the studied cities (Ulaanbaatar, Erdenet, and Darkhan). Typically, the HI value decreased in the order of As > Cr > Pb > Ni > Zn. As, Pb, and Cr exhibited higher values close to the safe level, while Zn and Cd were the lowest. 

In general, the concentration of heavy metals in soil samples taken from the three cities was estimated and showed that there was no adverse health effect for both cancer and non-cancer risks in terms of the assessment in this study. All samples were taken randomly from settlement areas, so some highly polluted sites may not have been sampled. However, this is not a significant issue because this study is assessing the urban environment in general. Human health risk assessment can be an important tool in determining heavy metal pollution and for distinguishing the exposure pathways of concern in an urban environment.

## 5. Conclusions

A total of 78 samples were taken from three cities and the concentration of the five elements, as well as the contamination and health risk assessment, were investigated in this study. As a result of the contamination assessment of heavy metals, there were some differences and similarities within this study of three major cities, where about 52% of the population live. In most cases, the heavy metal concentration of soil in the cities was higher than their background values in Ulaanbaatar and Erdenet, and the concentration of elements generally increased in the order of As < Cr < Ni < Pb < Zn. Sometimes the order of metals, except for Zn, fluctuated due to each city’s own geographical features.

The pollution level of the heavy metals for each city was estimated using I_geo_ and IPI. Ulaanbaatar and Erdenet cities were found to have a contamination level in the category of “uncontaminated to moderately contaminated.” Conversely, the I_geo_ mean value for all elements was lower than 0, or “uncontaminated,” in Darkhan. Regarding the pollution index, there were some differences in settlement areas depending on their socio-economic features and the aging of settlement areas. Level of IPI for functional areas in cities decreased in the order of industrial area > ger district > apartment district > common area.

Though the *HQ* and HI for both children and adults had differences between their values, the risk assessment for each metal showed the same trends. The *HQ* of ingestion for soil particles for all metals was much higher than those of inhalation and dermal absorption. The values of *HQ* and HI for those pathways of this study decreased in the order of ingestion > dermal contact > inhalation. For both children and adults, the *HQ*s and HIs of five metals were lower than the safe level (=1), indicating that there was little adverse health impact from them in the studied cities (Ulaanbaatar, Erdenet, and Darkhan) and the highest value was 4.15E-03 for Ulaanbaatar city. Commonly, the HI value decreased in the order of As > Cr > Pb > Ni > Zn. The HQ_ing_ values of arsenic had a major contribution to HI, however, its value (1.96E-04 to 4.00E-03 for children in studied cities) indicated lower than the safe level.

Levels of risks regarding cancer for As, Cr, and Ni were much lower than the tolerable range (10^−6^–10^−4^), above which the environmental and regulatory agencies perceive the risk to be unacceptable. In conclusion, this research is useful for both residents, in taking protective measures, and government, in alleviating heavy metals contamination, of the urban environment and urban perspective planning.

## Figures and Tables

**Figure 1 ijerph-16-02552-f001:**
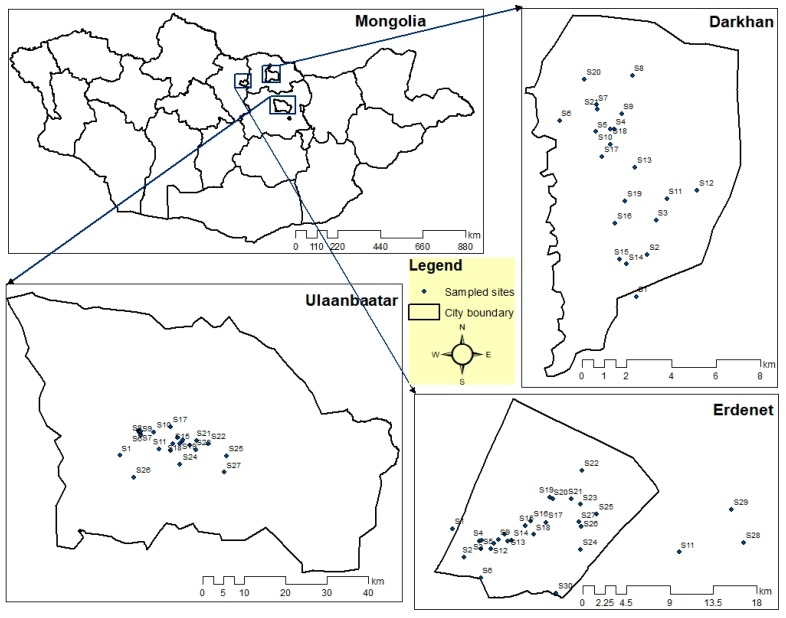
The position of samples taken from the study areas.

**Figure 2 ijerph-16-02552-f002:**
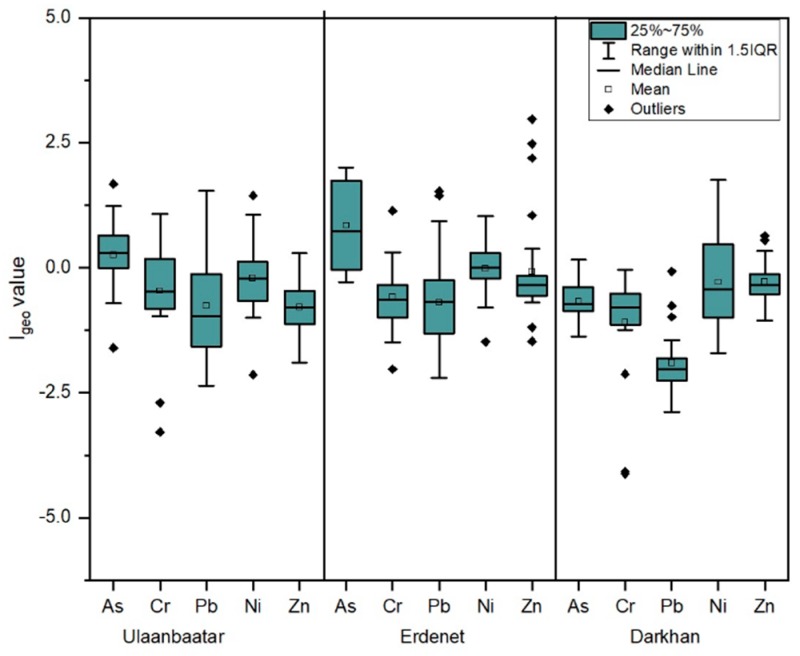
Boxplots of the I_geo_ values for the five heavy metals in soil of three cities (Ulaanbaatar, Erdenet, and Darkhan).

**Figure 3 ijerph-16-02552-f003:**
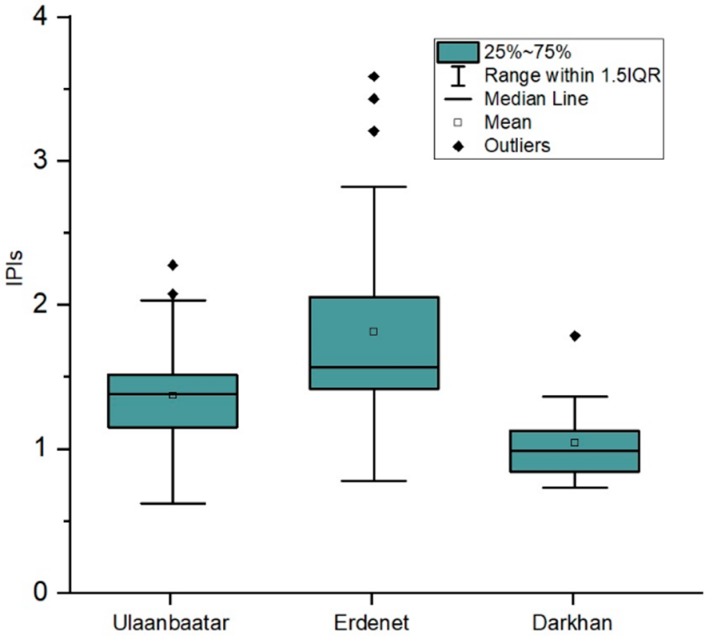
Boxplots of the integrated pollution index values for total sampling sites within this study from each city (Ulaanbaatar, Erdenet, and Darkhan).

**Table 1 ijerph-16-02552-t001:** Some information regarding the study area.

Parameters	UB	Erdenet	Darkhan
Period of settlement (year)	300	46	58
A total area of settlement zone (km^2^)	304.67	113.85	34.4
Number of sampled points (piece)	27	30	21
Amount of population (thousand)	1,462,973	99,758	85,378
The population per square kilometer in the settlement area (thousand)	4802	876	2482
Number of vehicles (thousand)	401,725	18,376	7799
Total households in ger district (thousand)	91,646	5778	3648

**Table 2 ijerph-16-02552-t002:** Classification of the geo-accumulation index.

Value	Soil Quality
I_geo_ ≤ 0	Uncontaminated
0 < I_geo_ < 1	Uncontaminated to moderately contaminated
1 < I_geo_ < 2	Moderately contaminated
2 < I_geo_ < 3	Moderately to heavily contaminated
3 < I_geo_ < 4	Heavily contaminated
4 < I_geo_ < 5	Heavily to extremely contaminated
5 < I_geo_	Extremely contaminated

**Table 3 ijerph-16-02552-t003:** Exposure factors for dose models that were used for this study.

Factor	Definition	Unit	Value			Reference
			Children		Adult	
*C*	concentration of an element in soil (UCL ^1^ 95%)	mg/kg	-		-	This study
*R_ing_*	ingestion rate of soil	mg/day	200		100	[35]
*EF*	exposure frequency	days/year	180		180	[34]
*ED*	exposure duration	years	6		24	[32]
*BW*	average body weight	kg	15		70	[35]
*AT*	average time	days	365 × *ED*		365 × *ED*	[35]
*CF*	conversion factor	kg/mg	1 × 10^−6^		1 × 10^−6^	[36]
*R_inh_*	inhalation rate	m^3^/day	7.6		20	[37]
*PEF*	particle emission factor	m^3^/kg	1.36 × 10^9^		1.36 × 10^9^	[34]
*SA*	the surface area of the skin that contacts the dust	cm^2^	2800		5700	[34]
*SL*	skin adherence factor for the dust	mg/cm^2^	0.07		0.2	[32]
*ABS*	dermal absorption factor (chemical specific)		0.03 for As, 0.001 for other metals			[34]
*AT*	average time carcinogenic	days	25,550	25,550		

^1^ UCL: Upper Confidence Level

**Table 4 ijerph-16-02552-t004:** Heavy metal mean concentration with their background values in the soil of three different cities from Mongolia.

City	Type of Indication	As	Cr	Pb	Ni	Zn
Ulaanbaatar	Concentration	28.04	16.56	43.11	21.26	106.11
	Background	14.17	13.04	37.91	14.74	114.10
Erdenet	Concentration	12.78	65.70	18.06	29.30	155.17
	Background	4.00	60.00	15.00	18.60	77.80
Darkhan	Concentration	3.33	31.90	20.90	19.49	67.26
	Background	3.33 ^1^	38.20	46.66	12.73	52.30
	MPC [38]	6.00	150.00	100.00	150.00	300.00

^1^ Owing to the background value of As being unable to determined, its average value was used for the contamination assessment.

**Table 5 ijerph-16-02552-t005:** Geo-accumulation index of heavy metals in urban soil in the major cities from Mongolia.

Elements	Geo-accumulation Index								
	Min.	Mean	Max.	Min.	Mean	Max.	Min.	Mean	Max.
	Ulaanbaatar			Erdenet			Darkhan		
As	−1.60	0.26	1.68	−0.29	0.84	2.00	−1.37	−0.67	0.16
Cr	−3.29	−0.46	1.07	−2.03	−0.59	1.14	−4.13	−1.10	−0.04
Pb	−2.37	−0.76	1.55	−2.20	−0.69	1.53	−2.89	−1.91	−0.08
Ni	−2.14	−0.21	1.44	−1.47	−0.08	2.97	−1.06	−0.28	0.64
Zn	−1.90	−0.78	0.29	−1.48	−0.02	1.03	−1.71	−0.29	1.75

**Table 6 ijerph-16-02552-t006:** Integrated pollution index (IPI) in different functional areas and each city (Ulaanbaatar, Erdenet, and Darkhan).

The List of Sampling Sites	Type of Functional Area	IPIs	Mean
Ulaanbaatar			
S2, S3, S4, S5, S6, S7, S8, S9	Industrial area	1.67	1.33
S1, S10, S11, S13, S14, S15, S16, S17, S18, S19, S20, S21, S22	Ger district	1.32	
S12, S23, S24, S25, S26, S27	Apartment district	1.0	
Erdenet			
S2, S3, S4, S6, S7, S15	Ger district	2.01	
S5, S10, S14, S19, S20	Apartment district	1.55	1.83
S1, S8, S9, S11, S12, S27, S28, S29, S30, S18, S23	Common area	1.54	
S13, S16, S17, S21, S22, S24, S25, S26	Industrial area	2.21	
Darkhan			
S3, S4, S5	Common area	0.76	0.94
S6, S9, S12	Ger district	0.87	
S7, S10, S11, S13	Apartment district	0.93	
S1, S2, S8, S14, S15, S16, S17, S18, S19, S20, S21	Industrial area	1.20	

**Table 7 ijerph-16-02552-t007:** HQs and HIs from heavy metals that given with their concentration and reference dose in the studied cities’ soil.

Elements	C (UCL 95%)	*RfD_ing_*^1^ mg/kg *d	*RfD_inh_*^2^ mg/kg *d	*RfD_derm_*^3^ mg/kg *d	*HQ_ing_^4^*			*HQ_inh_^5^*		*HQ_derm_^6^*		HI ^7^	
					Children		Adults	Children	Adults	Children	Adults	Children	Adults
Ulaanbaatar													
As	32.99	3.00 × 10^−4^	3.01 × 10^−4^	1.23 × 10^−4^	4.00 × 10^−3^	4.28 × 10^−4^		3.91 × 10^−5^	2.20 × 10^−5^	1.17 × 10^−4^	2.84 × 10^−4^	4.15 × 10^−3^	7.34 × 10^−4^
Cr	19.83	5.00 × 10^−3^	2.86 × 10^−5^	2.50 × 10^−4^	1.45 × 10^−4^	1.55 × 10^−5^		1.41 × 10^−6^	7.97 × 10^−7^	1.42 × 10^−7^	3.43 × 10^−7^	1.46 × 10^−4^	1.66 × 10^−5^
Pb	57.12	3.50 × 10^−3^	3.52 × 10^−3^	5.25 × 10^−4^	5.99 × 10^−4^	6.42 × 10^−5^		5.86 × 10^−6^	3.30 × 10^−6^	5.87 × 10^−7^	1.42 × 10^−6^	6.06 × 10^−4^	6.89 × 10^−5^
Ni	25.49	2.00 × 10^−2^	2.06 × 10^−2^	1.00 × 10^−3^	4.65 × 10^−5^	4.98 × 10^−6^		4.55 × 10^−7^	2.56 × 10^−7^	4.56 × 10^−8^	1.10 × 10^−7^	4.70 × 10^−5^	5.35 × 10^−6^
Zn	121.50	3.00 × 10^−1^	3.00 × 10^−1^	6.00 × 10^−2^	1.48 × 10^−5^	1.59 × 10^−6^		1.45 × 10^−7^	8.18 × 10^−8^	1.45 × 10^−8^	3.51 × 10^−8^	1.50 × 10^−5^	1.71 × 10^−6^
Erdenet													
As	12.78	3.00 × 10^−4^	3.01 × 10^−4^	1.23 × 10^−4^	1.88 × 10^−3^	2.01 × 10^−4^		1.84 × 10^−5^	1.04 × 10^−5^	5.52 × 10^−5^	1.34 × 10^−4^	1.95 × 10^−3^	3.45 × 10^−4^
Cr	65.70	5.00 × 10^−3^	2.86 × 10^−5^	2.50 × 10^−4^	5.68 × 10^−4^	6.08 × 10^−5^		5.55 × 10^−6^	3.13 × 10^−6^	5.56 × 10^−7^	1.34 × 10^−6^	5.74 × 10^−4^	6.53 × 10^−5^
Pb	18.06	3.50 × 10^−3^	3.52 × 10^−3^	5.25 × 10^−4^	2.47 × 10^−4^	2.64 × 10^−5^		2.41 × 10^−6^	1.36 × 10^−6^	2.42 × 10^−7^	5.84 × 10^−7^	2.49 × 10^−4^	2.84 × 10^−5^
Ni	29.30	2.00 × 10^−2^	2.06 × 10^−2^	1.00 × 10^−3^	6.05 × 10^−5^	6.48 × 10^−6^		5.92 × 10^−7^	3.34 × 10^−7^	5.93 × 10^−8^	1.43 × 10^−7^	6.11 × 10^−5^	6.96 × 10^−6^
Zn	155.17	3.00 × 10^−1^	3.00 × 10^−1^	6.00 × 10^−2^	2.75 × 10^−5^	2.95 × 10^−6^		2.69 × 10^−7^	1.52 × 10^−7^	2.70 × 10^−8^	6.52 × 10^−8^	2.78 × 10^−5^	3.17 × 10^−6^
Darkhan													
As	3.99	3.00 × 10^−4^	3.01 × 10^−4^	1.23 × 10^−4^	1.96 × 10^−4^	2.10 × 10^−5^		1.92 × 10^−6^	1.08 × 10^−6^	5.77 × 10^−6^	1.39 × 10^−5^	2.04 × 10^−4^	3.60 × 10^−5^
Cr	39.52	5.00 × 10^−3^	2.86 × 10^−5^	2.50 × 10^−4^	1.20 × 10^−4^	1.29 × 10^−5^		1.17 × 10^−6^	6.62 × 10^−7^	1.18 × 10^−7^	2.84 × 10^−7^	1.21 × 10^−4^	1.38 × 10^−5^
Pb	28.03	3.50 × 10^−3^	3.52 × 10^−3^	5.25 × 10^−4^	1.44 × 10^−4^	1.55 × 10^−5^		1.41 × 10^−6^	7.95 × 10^−7^	1.41 × 10^−7^	3.42 × 10^−7^	1.46 × 10^−4^	1.66 × 10^−5^
Ni	27.64	2.00 × 10^−2^	2.06 × 10^−2^	1.00 × 10^−3^	3.18 × 10^−5^	3.41 × 10^−6^		3.11 × 10^−7^	1.75 × 10^−7^	3.12 × 10^−8^	7.54 × 10^−8^	3.22 × 10^−5^	3.66 × 10^−6^
Zn	79.44	3.00 × 10^−1^	3.00 × 10^−1^	6.00 × 10^−2^	3.23 × 10^−6^	3.46 × 10^−7^		3.16 × 10^−8^	1.78 × 10^−8^	3.17 × 10^−9^	7.66 × 10^−9^	3.27 × 10^−6^	3.72 × 10^−7^

^1^ Reference dose—Ingestion. ^2^ Reference dose—Inhalation. ^3^ Reference dose—Dermal contact. ^4^ Hazard quotient—Ingestion. ^5^ Hazard quotient—Inhalation. ^6^ Hazard quotient—Dermal contact. ^7^ Hazard index.

**Table 8 ijerph-16-02552-t008:** Carcinogenic risks for metal elements in soil collected from major cities in Mongolia.

Heavy Metal Elements	As	Cr	Ni
Inh SF ^1^	15.1	42	0.0421
Risk—Ulaanbaatar	2.54 × 10^−10^	5.51 × 10^−11^	7.99 × 10^−9^
Risk—Erdenet	1.20 × 10^−10^	2.16 × 10^−10^	4.61 × 10^−9^
Risk—Darkhan	1.25 × 10^−11^	4.58 × 10^−10^	2.43 × 10^−9^

^1^ Inhalation Slope Factor.

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
