# Peer review of "Contamination and Health Risk Assessment of Heavy Metals in the Soil of Major Cities in Mongolia"

_ijerph, 2019, doi:10.3390/ijerph16142552_

Round 1

Reviewer 1 Report

General Comments:

This MS investigated five heavy metals in soils of three cities in Mongolia, and assessed the contamination levels and human health risk. The information obtained in this study would be helpful to understand the contamination situation of these metals in such cities, while a range of problems existed as indicated in the following parts, hence a minor revision is recommended.

Detailed Comments:

(1)Figure 1 on page 3, it is better to mark the latitude/longitude or some landmarks on the three city maps ;

(2)In section 2, how the soil samples were determined for the heavy metal concentration? And the quality control and assurance? Also, whats the soil depth when sample collected?

(3)Page 4, whats the equation of the IPI index ? namely, how it is calculated?

(4)Line 145/page 5, ... the values of ADD are calculated in Table 4 , while, Table 4 contains NO such information, please check; 

 (5)Line 169/Page 5, delete and Pb;  line 171/page 5, change five to three;

(6)Table 3 / page 6, does the symbol of C refer to the 95% UCL as mentioned in line 297/page 10 ; and check the spelling of the meaning of symbol of FE, dermal exposure ratia?

(7)Line 233-234/page 7, please check this sentence with Figure 2, since from the Figure 2, it should be the Igo value of Cr and Zn that below zero, not Cr and Ni;

(8)Line 307-309/page 10, change the sentence of ...As, Pb and Cr exhibited higher values close to safe level, while Zn and Cd are lowest to ...As, Pb and Cr exhibited higher values than that of Zn and Cd.  change the sentence of The value of As that is a major contribution to .... safe level to Arsenic has the highest HI value, indicating it is the major contributor to HI, but its HI still much lower than the safe level;

(9) Reference format (not completing) should be checked for Ref. 3, 4, 7, 11, 14, 16, 22, 27.

Author Response

Hello. We greeting to you and thanks for your comment. The feedback for the comment was written above list and revising procedure was performed on the original file. One emphasizing thing is background value of heavy metals that is used for estimation of Geoaccumulation Index and Integrated Pollution Index was changed as for Ulaanbaatar. We chose the background site and took soil samples. Then we used it for estimation in this study.

1.      According to the comment, it was performed

2.  According to the comment, sampling depth and chemical analyzing method were written. The certificate of assurance for Khanlab LLC attached the last page of this document with its translation.

3.      According to the comment, it was explained and equation was added in document.

4.      It is technical fault and was revised.

5.      It is technical fault and was revised.

6.      According to the comment, it was emphasized.

7.      According to the comment, it was revised.

8.      According to the comment, it was revised.

9.      According to the comment, it was revised.

MONGOLIA AGENCY FOR STANDARD AND METROLOGY

CERTIFICATE OF ASSURANCE

Number TL 82

The laboratory of Khanlab LLC
(Apartment of Khanrashaan, Street of Zaisan, Ulaanbaatar city)

which has complied with the requirement of ISO/IEC 17025:2017 (MNS ISO/IEC 17025:2018) standard has been assured to analyze the test.

This assurance proves that has the technical capability to perform the test within the defined scope and has a laboratory quality management system (based on the ISO-ILAC-IAF 2017 joint information).

Date of initially assured:                                              Date of assurance approved: 27.03.2019
20.11. 2012                                                                                      Expiration date: 16.02.2016

VICE HEAD OF MONGOLIAN AGENCY FOR STANDARD AND METROLOGY

Bilguun B.

Reviewer 2 Report

1.      Line 99. In Figure 1, some sample sites (s11, s28, and s29) are out the boundary of Erdenet City. Why to choose these sites? It should be explained.

2.      Line 110. How to analyze these heavy metals? What kind of procedure is applied for this concentration analysis? Why do only choose these 5 elements (As, Cr, Pb, Ni and Zn)? How about Mn (Manganese)?

3.      Line 195. In Table 4, it is significantly apparent that the mean concentrations of As, Pb and Ni in the soil of UB exceed their background values. However, the other metals Cr and Zn exhibit values below the background values. Why?

4.      Line 212. The mean concentrations of Cr and Pb are lower than their background values in Darkhan City while Ni and Zn are contrary. Why? Explain these phenomenon, please.

5.      Line 331. In Conclusion, the authors point out that “In most cases, the heavy metal concentration of soil in the cities is higher than their background values.” However, the Cr and Zn values are lower than background values in UB, and Cr and Pb are also lower. This conclusion is not correct. Maybe it needs to be modified.

6.      This paper just only shows the results of heavy metal concentrations in three cities. However, it is necessary to explain real reasons more detail that caused the higher values of heavy metal concentrations.

Author Response

Hello. We greeting to you and thanks for your comment. The feedback for the comment was written above list and revising procedure was performed on the original file. One emphasizing thing is background value of heavy metals that is used for estimation of Geoaccumulation Index and Integrated Pollution Index was changed as for Ulaanbaatar. We chose the background site and took soil samples. Then we used it for estimation in this study.

1.      This issue is a bit complicated due to the classification of prefecture. Actually, Erdenet city is located Bayan-Ondur soum (soum is a name of Mongolian prefecture). But the settlement area of this city is expanding gradually. Thus, there are citizens of Erdenet city in S11, S28, and S29.

2.      Most of scientific articles related to soil heavy metal study are usually chose those elements. We want to add Cadmium (Cd) to in this study, but its concentration is too low and wasn’t defined in laboratorial condition where chemical analyses were performed. As for Manganese, data on Darkhan city is available. Number of elements that was analyzed are limited in Ulaanbaatar and Erdenet.

3, 4, 5, 6. We revised some comments and need read and research more information about these issues. We hope that we can give proper explanation for those phenomenon by next round.

MONGOLIA AGENCY FOR STANDARD AND METROLOGY

CERTIFICATE OF ASSURANCE

Number TL 82

The laboratory of Khanlab LLC
(Apartment of Khanrashaan, Street of Zaisan, Ulaanbaatar city)

which has complied with the requirement of ISO/IEC 17025:2017 (MNS ISO/IEC 17025:2018) standard has been assured to analyze the test.

This assurance proves that has the technical capability to perform the test within the defined scope and has a laboratory quality management system (based on the ISO-ILAC-IAF 2017 joint information).

Date of initially assured:                                              Date of assurance approved: 27.03.2019
20.11. 2012                                                                                      Expiration date: 16.02.2016

VICE HEAD OF MONGOLIAN AGENCY FOR STANDARD AND METROLOGY

Bilguun B.

Round 2

Reviewer 2 Report

This manuscript had revised well. It is possible for publish.